# Optimization Study for the Desorption of Methylene Blue Dye from Clay Based Adsorbent Coating

**Momina [1]**, **Mohd Rafatullah [2],\***, **Suzylawati Ismail [1],\*** and **Anees Ahmad [3]**

1 School of Chemical Engineering, Universiti Sains Malaysia, Engineering Campus, Nibong Tebal 14300, Penang, Malaysia; momina.anees@gmail.com
2 School of Industrial Technology, Universiti Sains Malaysia, Main Campus, Gelugor 11800, Penang, Malaysia
3 Department of Industrial Chemistry, Aligarh Muslim University, Aligarh 202001, UP, India; aneeschem@gmail.com
* Correspondence: mohd_rafatullah@yahoo.co.in or mrafatullah@usm.my (M.R.); chsuzy@usm.my (S.I.); Tel.: +604-653-2111 (M.R.); Fax: +604-653-6375 (M.R.)

**Abstract:** Batch desorption experiments of methylene blue (MB) dye from a clay adsorbent coating were carried out to evaluate the maximum desorption conditions. Combination of thermal and chemical regeneration techniques were used for the desorption process. The desorption of MB was found to be 70% using an HCl solvent after heating adsorbent coating at 160 °C. The optimization study was carried out to identity the optimum desorption conditions using MINITAB 14 software. The individual and interaction effects of three factors, temperature, dye concentration and contact time for desorption of dye were determine by applying response surface methodology (RSM). The optimization results showed that all three factors have main effects whereas the interaction of concentration–time is significant as compared to other interactions. The findings exhibit a maximum desorption efficiency 23 mg/g at 60 °C for 100 mg/L of dye and 150 min of contact time.

**Keywords:** clay; desorption; methylene blue; optimization; response surface methodology

---

## 1. Introduction

We are living in challenging times where there have been substantial changes in the climate pattern of the planet due to excessive and irresponsible industrial and individual activities. Industries like textile industries consumes a large amount of water and generate almost 90% of wastewater [1]. Different types of dyes are being used by textile industries for coloring fabrics in large numbers, which are disposed into water bodies. Huge amount of wastes produced by textile manufacturers can have a serious environmental impact on the quality of water [2–4]. Adsorption is one of the superior processes and wide acceptability in the removal of organic dyestuff from water. Due to the high efficiency, simplicity and flexibility of design, ease of operation and non-toxicity of the utilized adsorbents, the adsorption process is strongly favored over other different technology for the treatment of wastewater [5–9]. Activated carbon (AC) is widely used as an adsorbent for the removal of dyes. However, due to its typical high cost, the use of AC is restricted, and the developed techniques are quite expensive for its regeneration [10,11].

Recently, clay-based adsorbents, accomplished an important rank in environmental engineering applications especially wastewater treatment due to their high adsorption capacity, low cost and abundant availability [12]. Among clay-based adsorbents, bentonite is the most commonly used adsorbents for water purification. It has a great affinity towards the cationic dye due to the attraction of opposite charges on the surface of the lattice [13,14]. An adsorbent coating is a new approach of adsorbent where the ordinary form of adsorbent has been reformulating and transforms into a

liquid/slurry form, which has then been laminated onto inert surfaces or substrates. Bentonite coating is an efficient methodology not only for the durability and low cost, but also due to its acceptability in industries. Thin coated bentonite-based adsorbent had been prepared by mixing bentonite, water-based binder and solvent in a specific ratio to remove methylene blue (MB) from synthetic dye solution, achieving a higher adsorption capacity (213.57 mg/g) [15] as compared to adsorption of MB on bentonite clay alone (175 mg/g) [5]. For a few reasons, it is desirable to have the solid adsorbent deposited on a substrate as a coating instead of being contained in particulate form as pellets, beads, flakes, particles, powder, etc. Adsorbent coatings improve the catalytic and adsorption capacity of adsorbents by increasing the surface area/weight ratio, reduces the quantity of solid adsorbent required, enhances the binding strength, protects the substrate from harmful environment and performs a specific desorptive or catalytic role over the entire surface of the substrate [16].

However, the problem of dyes loaded adsorbent is also challenging because the continuous accumulation of pollutants on the adsorbents gradually reduces the overall adsorption efficiency of the adsorbent and also affects the environment due to leaching of adsorbate [17]. Regeneration of adsorbents is important, using a proper technique helps in improving adsorption efficiency by removing pollutants, improves the overall cost of treatment process, reduces the generated waste and tackles the disposal problem. According to literature reviews, the prevalent and widely used regeneration techniques for clay based adsorbents are chemical, thermal, supercritical extraction, photocatalytic and biological techniques [18]. This study is based on the regeneration of bentonite based adsorbent coating using the combined method of thermal and chemical regeneration. Experimental design analysis is an important technique because it provides the statistical model that helps in optimizing different parameters and their interaction [19,20]. Moreover, it requires less number of experiments, time saving, minimizes amount of chemical requires and manpower [21]. Hence, this study focused on statistical optimization of process conditions for desorption of MB from adsorbent coating. Statistical optimization was carried out to determine the optimum conditions for desorption of MB. Moreover, the adsorbent coating was also characterized by surface electron microscope (SEM) after desorption of MB dye.

## 2. Materials and Methods

### 2.1. Chemicals and Solution Used

Methylene blue (MB) dye supplied from Merck Chemicals Malaysia was used as an adsorbate. Bentonite clay was used without any modification and supplied from Modern Lab Sdn. Bhd, Malaysia. Solvents such as ethanol and HCl obtained from R&M chemicals Pdn. Malaysia were used for desorption study. HCl and NaOH were used to adjust the pH of the solution, which was measured using a pH meter (Elico, Hyderabad, India). A water based binder was used as a support, which contains polyvinyl alcohol (PVA), ammonia, calcium carbonate, formalin and hydrosol. The contents present in the binder were similar to bentonite clay except for $SiO_2$, which were present in half amount to that of $SiO_2$ present in bentonite. The percentage of calcium carbonate was higher in the binder and also considered as a major content [15].

### 2.2. Adsorption Study

Batch adsorption study was performed in a similar manner as performed by Azha et al [15]. The adsorbent coating was prepared by the mixture of binder (2.5 g), bentonite (0.3 g) and distilled water (3 g). The mixed slurry was coated on the glass inner side wall of the beaker and oven dried at 80 °C for seven hours. The adsorbent coating was dipped in 200 mL MB solution with a different initial dye concentration that ranged between 50–100 mg/L at 250 rpm under room temperature 30 °C. The concentration of MB at equilibrium was measured using a UV-vis spectrophotometer (HASH UV-vis model DR6000 spectrophotometer, Loveland, CO, USA) at a maximum wavelength of 664 nm.

The adsorption capacity and adsorption removal were calculated using the following Equations (1) and (2):

$$q_{e,sorption} \text{ (mg/g)} = V(C_i - C_e)/M, \tag{1}$$

$$q\% = (C_i - C_e)/C_i \times 100, \tag{2}$$

where V is the volume of the dye solution (L), M is the mass of the adsorbent used (g), $C_e$ is the final concentration of MB (mg/L) and $C_i$ is the initial concentration of MB (mg/L). The adsorbent coating loaded with MB was dried in an oven at 70 °C for 4 h and further used for the desorption study.

## 2.3. Desorption Study

Desorption of MB from the adsorbent coating was performed using thermal and chemical regeneration methods. The spent adsorbent coating was heated at 160 °C for 45 min. This heated adsorbent coating further dipped in solvents at 30–60 °C. The solvents used in this study were HCl, ethanol and a different mixture of the HCl:ethanol ratio (75:25, 25:75 and 50:50). The selection of solvents was based on the desorption efficiency of the dye using different solvents. The concentration of MB desorbed in solvents was measured using a UV-vis spectrophotometer (HASH UV-vis model DR6000 spectrophotometer, Loveland, CO, USA). The desorption efficiency and desorption removal of the dye using both the thermal ($Q_H$; Equation (4)) and chemical regeneration ($Q_C$; Equation (5)) method are given below:

$$Q_{e,desorp} \text{ (mg/g)} = V(C_f/M), \tag{3}$$

$$Q_H\% = (W_{AA} - W_{AR})/(W_{AA} - W_{BA}) \times 100, \tag{4}$$

$$Q_C\% = (Q_{e,\, desorp}/q_{e,sorption}) \times 100, \tag{5}$$

where M is the weight of spent adsorbent (g), V is the volume of the solvent (L), $C_f$ is the MB concentration in the solvent (mg/L), $W_{AA}$ is the weight after adsorption (g), $W_{AR}$ is the weight after heating (g) and $W_{AB}$ is the weight before adsorption (g). The desorption efficiency is defined as the amount of dye desorbed from per gram of spent adsorbent at equilibrium.

## 2.4. Optimization Study

The optimal design to study the desorption of MB from adsorbent coating was analyzed using the MINITAB software. Optimization of effective parameters on the desorption process, i.e., temperature ($x_1$), concentration ($x_2$) and contact time ($x_3$) and their interaction was studied using response surface methodology (RSM). RSM is a statistical method that uses quantitative data from appropriate experiments to determine regression model equations and operating conditions. The optimization process involves three process; conducting a statistically designed experiment, determining mathematical model coefficients and prediction of the response value and to check the adequacy of the developed model. The range and variable of the experiment is given in Table 1.

**Table 1.** Range and levels used for the batch desorption study.

| Factors | Low Level (–1) | High Level (+1) |
|---|---|---|
| ($x_1$) Temperature (°C) | 30 | 60 |
| ($x_2$) Concentration (mg/L) | 50 | 100 |
| ($x_3$) Contact time (min) | 15 | 150 |

The general form for the desorption of MB from the clay adsorbent coating can be written as a linear or quadratic equation.

$$Y = \beta_o + \beta_1 x_1 + \beta_2 x_2 + \beta_3 x_3, \tag{6}$$

$$Y = \beta_o + \beta_1 x_1 + \beta_2 x_2 + \beta_3 x_3 + \beta_{11} x_1^2 + \beta_{22} x_2^2 + \beta_{33} x_3^2 + \beta_{12} x_1 x_2 + \beta_{13} x_1 x_3 + \beta_{23} x_2 x_3, \tag{7}$$

where Y is the response (desorption efficiency), $x_1$, $x_2$ and $x_3$ are the coded values of three variables and $\beta_o$, $\beta_1$, $\beta_2$, $\beta_3$, $\beta_{11}$, $\beta_{22}$, $\beta_{32}$, $\beta_{12}$, $\beta_{13}$ and $\beta_{23}$ are the regression coefficient constants of the developed model, respectively. Fitting of the polynomial equation was determined using the $R^2$ value. The model terms were also determined from *p*-values with a 95% confidence level. The optimization of factors and their interaction was carried out using a main effect, interaction, Pareto charts, surface and contour plots.

### 2.5. Characterization

The MB-loaded adsorbent coating was characterized by thermogravimetric analysis (TGA/DTA) using STA 6000 Thermogravimetric Analyzer (PerkinElmer, Waltham, MA, United States) under nitrogen atmosphere with a heating rate of 10 °C per minute. The regenerated clay adsorbent coating was characterized by scanning electron microscopy (SEM).

## 3. Results and Discussion

### 3.1. Desorption Study

In this study, a combination of two regeneration techniques were used, thermal regeneration followed by chemical regeneration (named as thermo–chem). The thermal stability of the binder and adsorbent coating before and after adsorption of MB was performed using TGA/DTA analysis. The TGA/DTA curve are shown in Figure 1.

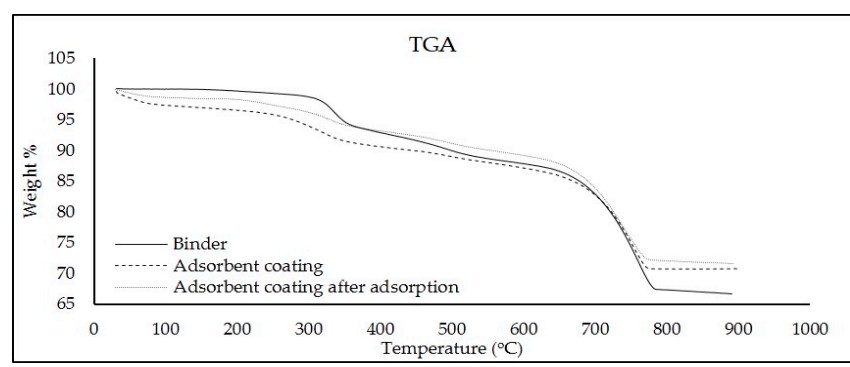

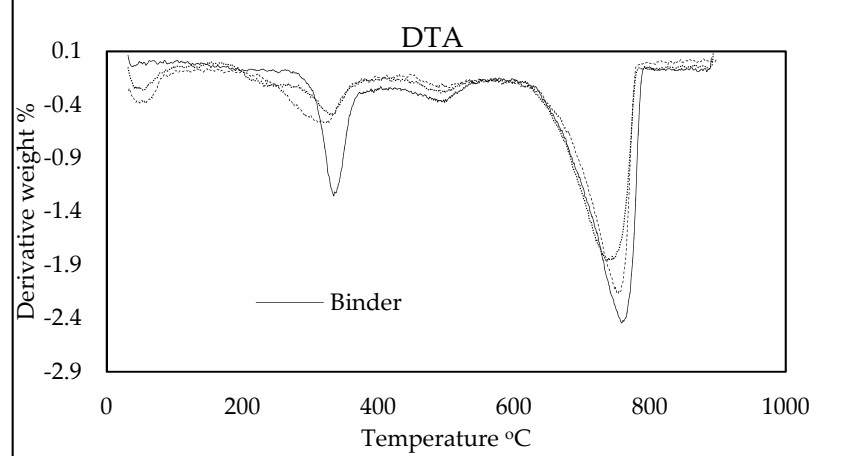

**Figure 1.** TGA/DTA analysis.

Since the boiling point of MB is 110 °C, there was no peak that appeared at that temperature. Since the interaction between MB and the clay based adsorbent coating is an electrostatic interaction [15], which thereby fails to desorb the dye at 110 °C. However, the peak at 200–230 °C was observed, which was not available either in the binder or before adsorption. This peak shows the desorption of MB

due to the metabolite formation of MB, which decomposes at this temperature. Moreover, heating of the spent adsorbent coating at 200 °C turns brown in color, which may be due to the presence of the binder in the mixture of the adsorbent. Therefore, for further experiments the heating temperature was used in between 150–200 °C. Moreover, the peaks appearing at 300 °C in every sample shows that the binder and adsorbent coating was stable and a further increase in temperature decreased the thermal stability. The desorption efficiency (calculated from Equation (4)) of the dye at different temperature is given in Table 2.

**Table 2.** Desorption efficiency of the dye at different heating temperatures.

| Heating Temperature (°C) | Desorption Efficiency (%) |
|:---:|:---:|
| 150 | 8.03 ± 1.07 |
| 160 | 8.14 ± 0.5 |
| 170 | 8.39 ± 0.67 |
| 190 | 8.99 ± 1.12 |

After heating the clay adsorbent coating at different solvents were tested for the further desorption of the dye. The highest efficiency achieved for desorption of MB was 70% with HCl as compared to pure ethanol at 60 °C. The mixtures of the solvents also affect the dye desorption efficiency of the adsorbent. According to Greluk and Hubicki, [22] only alcohol does not affect too much of the desorption of the dye while mixture of the acid to alcohol ratio can significantly improve the performance of the desorption process. In the present study desorption efficiencies of MB were performed with a 75:25, 50:50 and 25:75 ratio of acid to alcohol at 60 °C for both chemical regeneration and the combination of thermal and chemical regeneration. The desorption efficiency of the dye using the chemical method for the acid to alcohol ratio was found to be lower than pure ethanol as shown in Figure 2. The maximum desorption efficiency of 23% was attained for ethanol, which established a strong electrostatic attraction of MB with the adsorbent coating and requires higher energy to remove dye in the chemical regeneration method. Besides that, the dye desorption efficiency was found to be increased as compared to pure alcohol in the case of the combined method as shown in Figure 2. The desorption efficiency for the acid to alcohol ratio of 75:25 was 60% for the thermo–chem method. However, the desorption efficiency with pure alcohol was lower than the acid to alcohol ratio, which shows that the interaction between the MB dye and adsorbent is due to the electrostatic attraction, H-bonding and hydrophobic interaction [23].

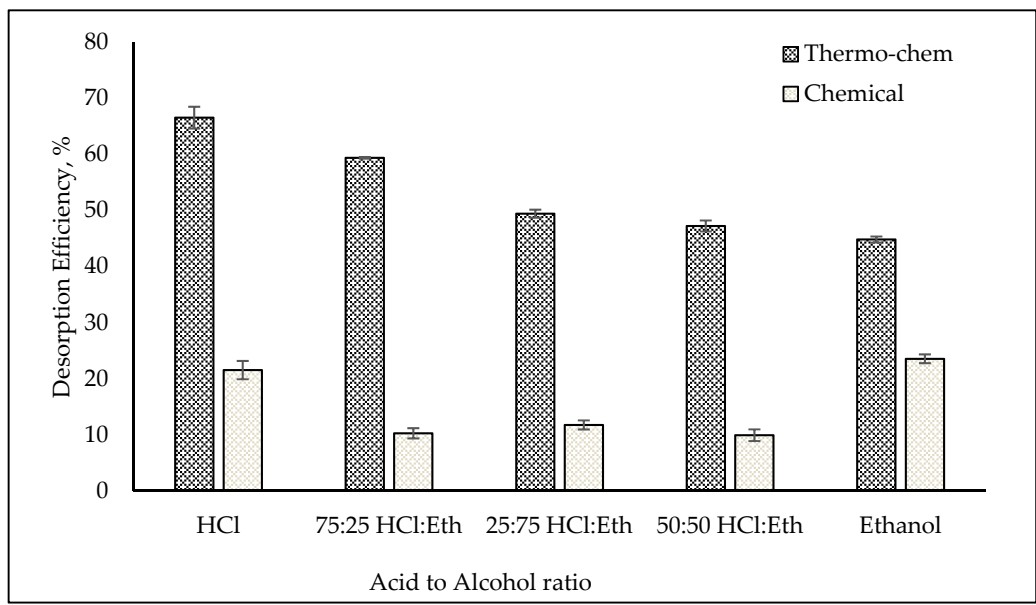

**Figure 2.** Effect of solvents on the desorption of methylene blue dye (MB) from the clay adsorbent coating.

*3.2. Optimization Study*

The design matrix of the coded and real values for the desorption of MB from the adsorbent coating are given in Table 3. The results obtained from the design matrix were used to evaluate the relationship between temperature ($x_1$), concentration ($x_2$) and contact time ($x_3$). The polynomial first order and interactive regression model equation was developed using the Minitab software. Therefore, model equations are shown in Equations (8) and (9).

$$y = -8.00 + 0.339x_1 + 0.108x_2 + 0.0574x_3 \ (R^2 = 0.973, p = 0.001), \tag{8}$$

$$y = -10.0 + 0.247x_1 + 0.171x_2 + 0.0062x_3 - 0.000060x_2x_3, \tag{9}$$

$$(R^2 = 0.982, p = 0.004).$$

The *p*-values and regression coefficient values of equation 8 and 9 demonstrated that the first order equation with one in the interactive term fits the experimental results better. All the linear terms in Equations (8) and (9) are significant as demonstrated in Table 4 due to *p*-values <0.05 with a 95% confidence level. However, the quadratic ($x_1^2$, $x_2^2$, $x_3^2$) and interactive terms ($x_1x_2$, $x_1x_3$) were insignificant ($p > 0.05$) and therefore removed from equation 7. This model describes that the concentration of MB ($x_2$) had a great effect on the desorption efficiency of the dye followed by the temperature ($x_1$), time ($x_3$) and concentration–time interaction ($x_2.x_3$). According to Equation (9), the interaction of concentration–time ($x_2.x_3$) had a negative effect on the desorption efficiency whereas temperature ($x_1$), concentration ($x_2$) and time ($x_3$) had a positive effect. The ANOVA table of the fitted model (Table 5) shows that the regression was with a $p = 0.004$, which thereby, confirms that the data model fits the experimental data.

**Table 3.** Experimental and predicted results from the polynomial regression analysis.

| Experiment | Coded Values | | | Real Values | | | Desorption Efficiency (mg/g) | Predicted Values |
|---|---|---|---|---|---|---|---|---|
| | $x_1$ | $x_2$ | $x_3$ | $x_1$ | $x_2$ | $x_3$ | | |
| 1 | 1 | −1 | −1 | 60 | 30 | 15 | 10.293 | 10.016 |
| 2 | −1 | −1 | −1 | 30 | 30 | 15 | 2.302 | 2.606 |
| 3 | 1 | −1 | 1 | 60 | 30 | 150 | 10.493 | 10.610 |
| 4 | −1 | −1 | 1 | 30 | 30 | 150 | 3.293 | 3.200 |
| 5 | 1 | 1 | −1 | 60 | 100 | 15 | 23.169 | 21.923 |
| 6 | 1 | 1 | 1 | 60 | 100 | 150 | 20.560 | 21.950 |
| 7 | −1 | 1 | 1 | 30 | 100 | 150 | 15.965 | 14.540 |

**Table 4.** Estimated regression coefficient for the desorption of MB.

| Predictor | Coef | SE Coef | T | *p* |
|---|---|---|---|---|
| Constant | −10.031 | 2.499 | −4.10 | 0.026 |
| $x_1$ | 0.247 | 0.0364 | 6.78 | 0.007 |
| $x_2$ | 0.171 | 0.0246 | 6.96 | 0.006 |
| $x_3$ | 0.0062 | 0.017 | 0.36 | 0.007 |
| $x_2 - x_3$ | −0.000060 | 0.00023 | −0.26 | 0.011 |

**Table 5.** ANOVA for desorption of MB.

| Source | Seq SS | DF | Adj SS | Adj MS | F Value | *p* Value |
|---|---|---|---|---|---|---|
| Main effects | 381.740 | 3 | 381.740 | 127.247 | 53.29 | 0.004 |
| Two-way interactions | 0.163 | 1 | 0.163 | 0.163 | 0.07 | 0.008 |
| Regression | 381.903 | 4 | 381.903 | 95.476 | 39.99 | 0.006 |
| Residual error | 7.163 | 3 | 7.163 | 2.388 | | |
| Total | 389.066 | 11 | | | | |

The predicted vs. observed value plot (shown in Figure 3a) shows that the experimental values were lying near to the straight line (R = 0.97) and shows a satisfactory correlation between the predicted and observed experimental values. However, the adequacy of the model was described by the standardized residual plots (shown in Figure 3b), which shows the difference between the actual response value and the value that fitted best under the hypothesized model [24]. The small residual values showed the accuracy of the model and there was no abnormality in this study.

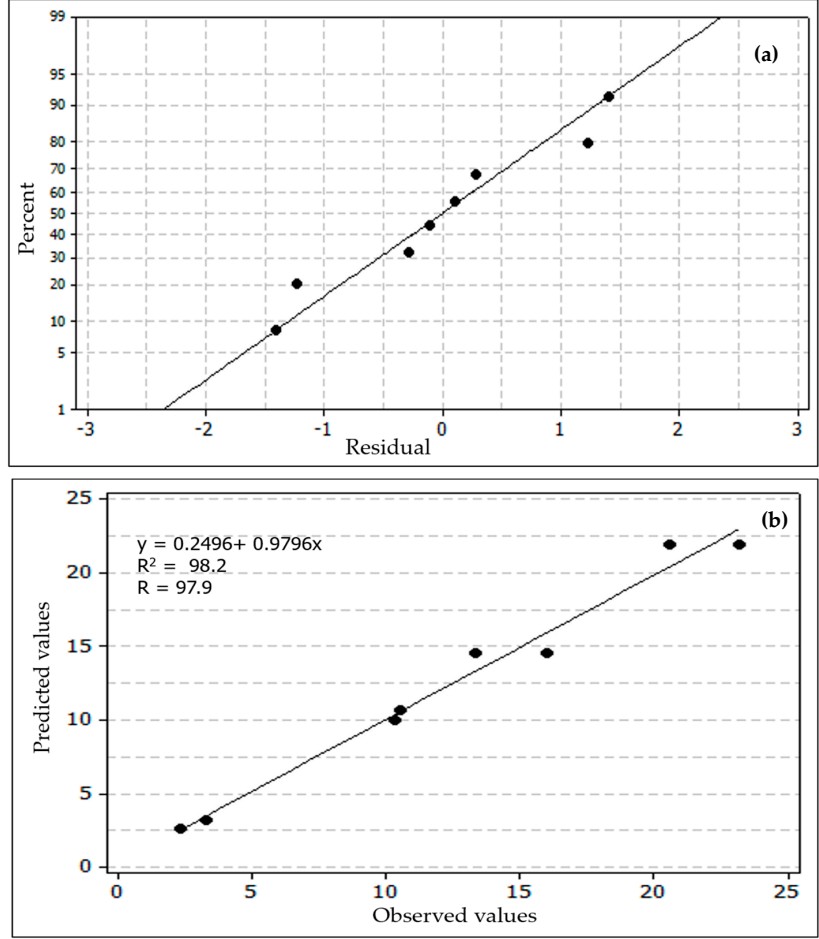

**Figure 3.** (**a**) Predicted values vs. the experimental values for the desorption of MB from the clay adsorbent coating while (**b**) Normal probability plot of the standardized residual.

### 3.2.1. Main Effects

The main effect of all three parameters for the desorption of MB are shown in Figure 3. The results of the regression analysis are represented by the main effect plot. The parameters, which are significant at a 95% confidence level, are represented in this plot. The main effects represent deviations of the average between the high and low levels for each factor. When the effect of a factor is positive, Qe increases as the factor changes from low to high levels. In contrast, if the effects are negative, a reduction in (Qe) occurs for a high level of the same factor [25]. The larger the change in value from −1 to 1, larger will be the desorption efficiency (Qe), which means that the significance of the factor depends on the length of the vertical line shown in Figure 4. The effect of three parameters; temperature, concentration and time is positive that is value of Qe increases with an increase in value from low to high values. Contact time ($x_3$) between the solvent (HCl) and adsorbent coating had a great effect on the desorption of the dye, as is evident by the longer vertical line. However, the temperature ($x_1$) of the solvent also plays an important role in the desorption process. Since the interaction between the dye and adsorbate coating is an electrostatic interaction (strongly bonded), therefore the dye cannot be able to desorb at a

lower temperature, which resulted in a lower desorption efficiency. Whereas, the desorption efficiency at a higher temperature increases, which showed that the solubility and distribution coefficient of MB in the solvent increased. Moreover, increment in the desorption efficiency with respect to the concentration ($x_2$) showed a high driving force for the mass transfer from the solid to liquid phase. Furthermore, the dye molecules tended to form an aggregate during the adsorption process, which resulted in a higher desorption efficiency.

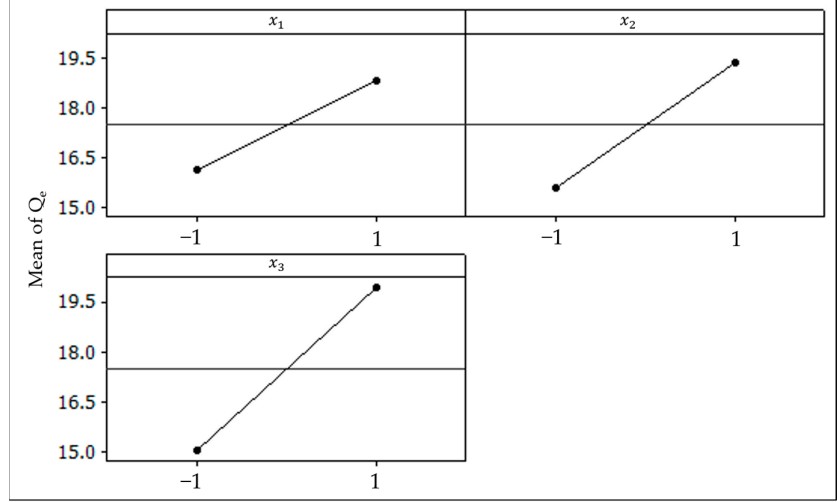

**Figure 4.** Main effects plots (of temperature, concentration and time) for Qe for the desorption of MB from the clay adsorbent coating.

### 3.2.2. Interaction Plots

These plots represent the significance of the interaction of two factors means that the change in response from the low to high level of one factor is dependent on the level of the second factor (i.e., lines are not parallel) [26]. The effect of the interaction of factors ($x_1.x_2$, $x_1.x_3$, $x_2.x_3$) is shown in Figure 5. The interaction of concentration and time ($x_2.x_3$) were found to be statistically significant for the determination of Qe as compared to the temperature–concentration and temperature–time interaction. The effect of the interaction of concentration–time was found to be more significant at a lower temperature. However, the temperature–concentration and temperature–time interaction were more significant at a higher temperature and resulted in higher desorption efficiency (Qe).

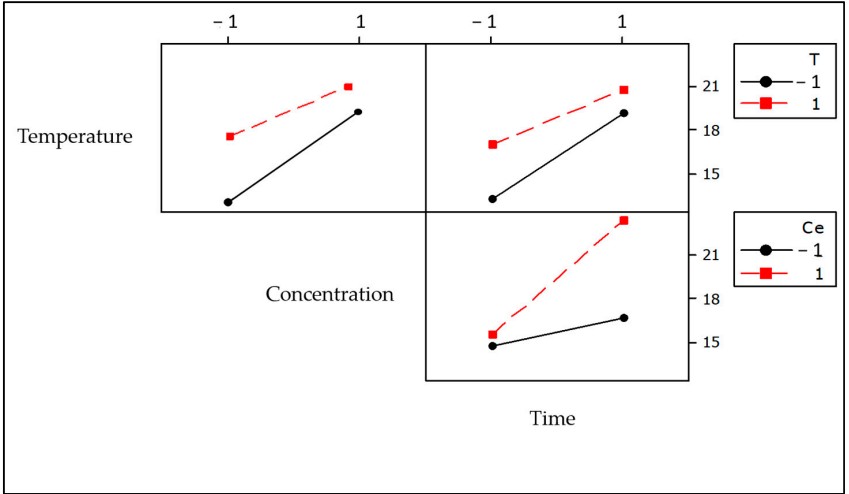

**Figure 5.** Interaction plots (of temperature–concentration, temperature–contact time, concentration–contact time) for the desorption of MB from the clay adsorbent coating.

### 3.2.3. The Pareto Chart

The relative importance of the main effects and interaction effects can be determined through the Pareto chart as shown in Figure 6. A student's *t*-test was performed to determine whether the calculated effects were significantly different from zero, these values for each effect are shown in the Pareto chart by horizontal columns [25]. For eleven degrees of freedom and a 95% confidence level the *t*-value was 1.78. The values that were exceeding the reference line were significant with a 95% confidence level whereas the values before the reference line were not significant. As shown in Figure 5, the main factors (temperature ($x_1$), time ($x_3$) and concentration ($x_2$)) and the interaction of concentration–time ($x_2x_3$) was found to be more significant at the 0.05 level. Whereas temperature–time ($x_1x_3$), temperature–concentration ($x_1x_2$) and temperature–concentration–time ($x_1x_2x_3$) had a smaller effect and were not significant for desorption of MB from the adsorbent coating.

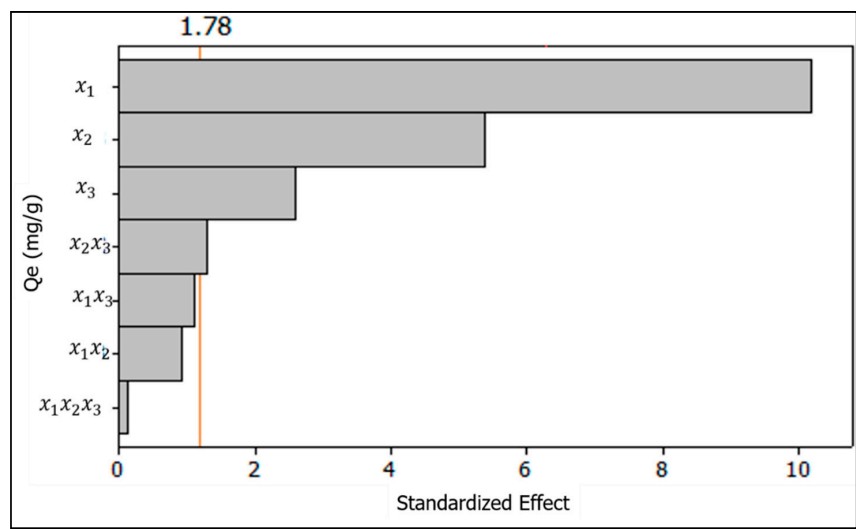

**Figure 6.** Pareto chart (*t*-test) for the desorption of MB from the clay adsorbent coating.

### 3.2.4. Contour and Surface Plots

The contour plots and response surface plots for the desorption of MB from the adsorbent coating are shown in Figures 7 and 8. These plots are generated by keeping one parameter at the optimum level and varying other parameters to study their main and interaction effect. The response surface plots show the estimated value of Qe as a function of the independent variable, the height of the surface represents the value of Qe. The surface and contour plots representing the same results as observed in the interaction plots. The curved lines appearing in the contour curves represent that the model contains the factor of the interaction of concentration–time. These graphs show that a higher desorption efficiency (23 mg/g) achieved at temperature 50–60 °C with an initial dye concentration of 50–100 mg/L for 60–150 min of contact time.

### 3.3. Characterization

The behavior and surface of the adsorbent before and after the desorption of MB using the thermo–chem method was analyzed using SEM images as shown in Figure 9. The adsorbent had a porous and flake-like structure before adsorption, which became smooth after adsorption of MB. The smoothness in the surface was due to the filling of the space or pores by MB molecules, which were available before adsorption. However, after desorption of MB using the thermo–chem method the roughness of adsorbent surface increased. The roughness showed the removal of MB molecules from the surface and pores of the adsorbent, which means that the surface was approaching towards an un-adsorbed type of the adsorbent. In conclusion, the adsorbent was not completely but partially regenerated with a desorption efficiency of 70% using HCl after heating at 160 °C. However, this was

a considerable increase in desorption with the thermo–chem method as compared to the chemical regeneration alone. This increment in the desorption efficiency of MB helps the regenerated adsorbent to further be used for more than five cycles of adsorption/desorption. This justifies the usage of this adsorbent for industrial purposes as more regeneration capability will reduce the cost of the process considerably.

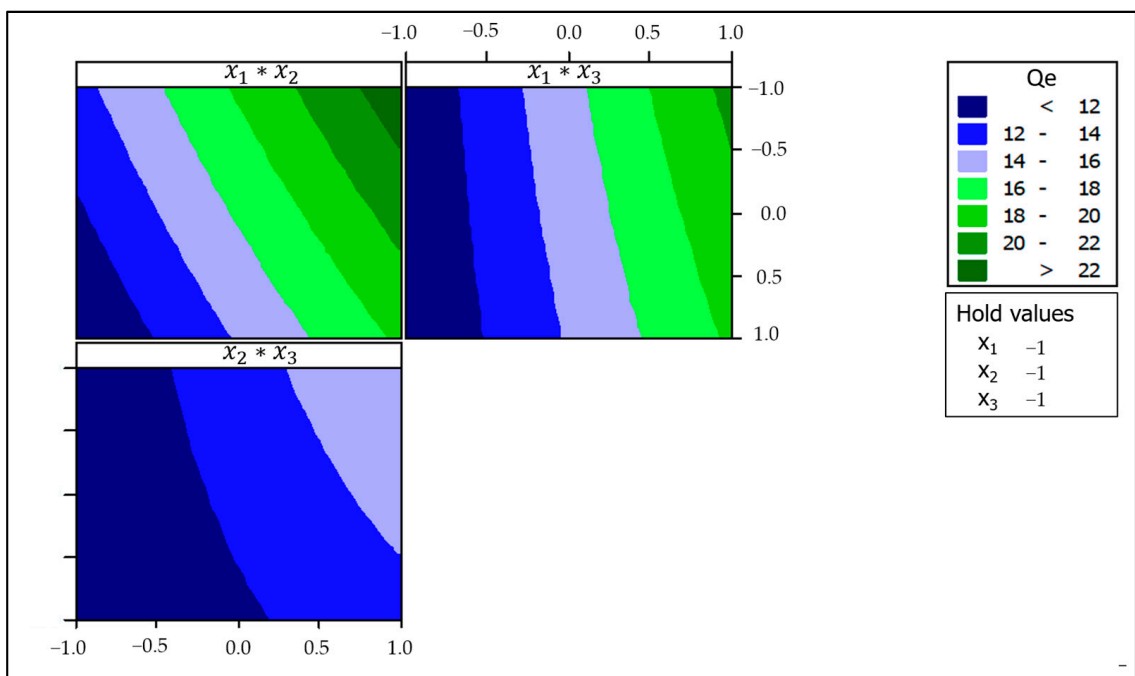

**Figure 7.** Contour plots for desorption of MB from the clay adsorbent coating.

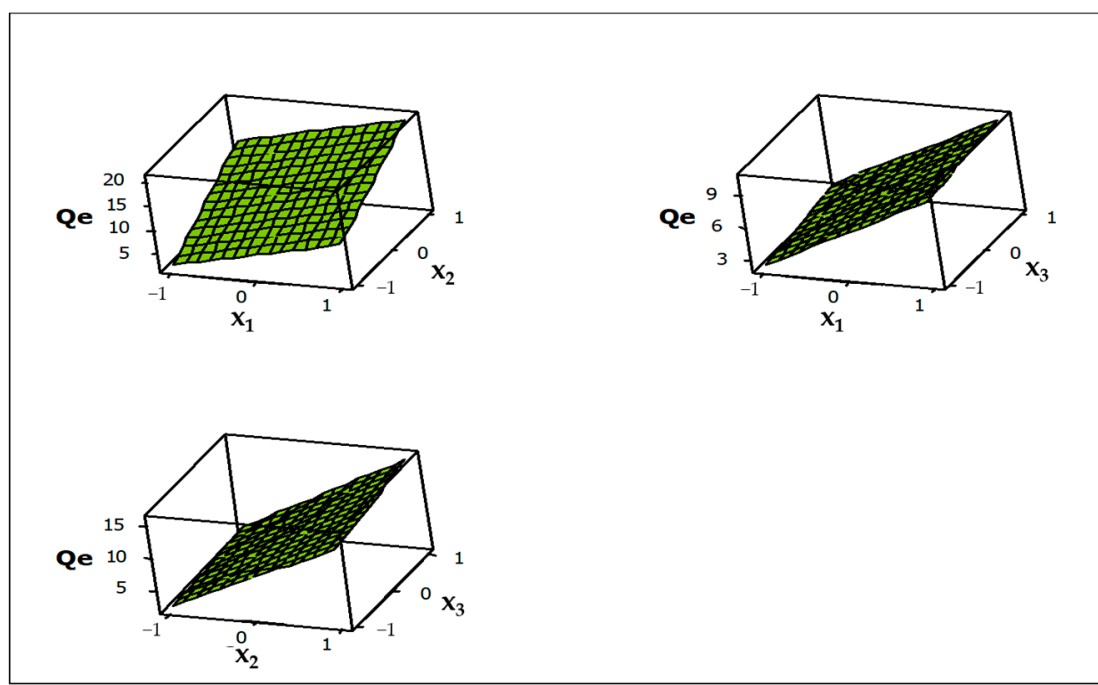

**Figure 8.** Response surface plots for desorption of MB from the clay adsorbent coating.

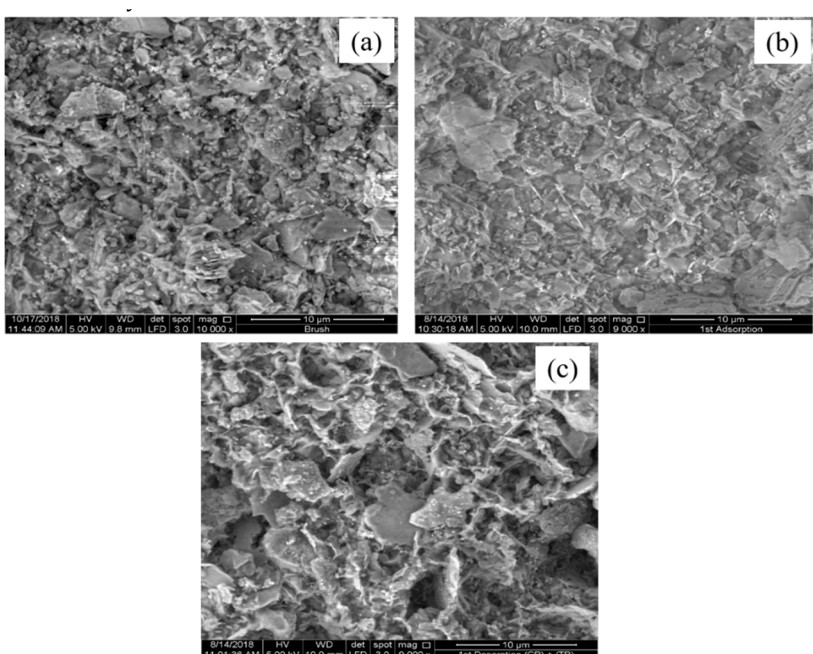

**Figure 9.** Scanning electron micrograph of the (**a**) adsorbent before adsorption, (**b**) adsorbent after adsorption of MB and (**c**) adsorbent after desorption of MB.

### 3.4. Desorption Mechanism

The possible desorption mechanism as indicated by the model is that it is a combined effect of the concentration, temperature and contact time. The temperature destabilizes adsorptive forces followed by reorientation of the molecules and then the kinetic energy. The solvating power of the solvent and concentration gradient helps in the diffusion or migration of MB molecules from the solid to liquid phase. In other words, the extension of the desorption is proportional to the temperature and concentration followed by the exposure time of the adsorbed molecules to the equilibrating solutions. In this way all the three factors affect the desorption of the adsorbed molecules. Another factor might be the surface area of the adsorbent, i.e., the higher the surface area the faster/more the desorption. In other words, desorption will be inversely proportional to particle size of the adsorbent, because the surface area decreased with an increase in particle size. The scheme for the desorption of adsorbed MB is shown in Figure 10.

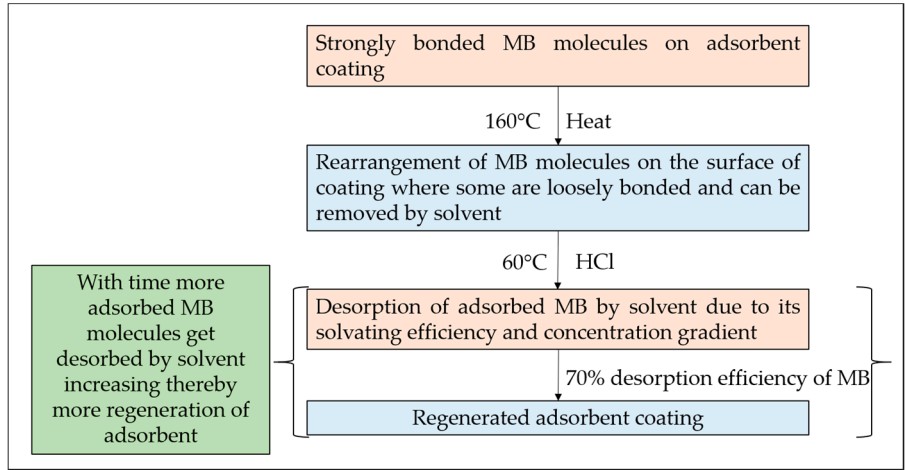

**Figure 10.** Mechanism scheme for the desorption of the adsorbed MB.

## 4. Conclusions

The desorption of the dye was performed using a combined method of thermal and chemical regeneration with a 70% desorption efficiency using HCl after heating the adsorbent coating at 160 °C. The optimization of all the parameters including temperature, time and concentration was also performed with techniques of regression. The statistical analysis demonstrated that the concentration, temperature and time had an individual effect on the desorption of MB. Moreover, a significant interaction was also observed between concentration and time. The factorial experiments demonstrated a significant antagonistic interaction between pH and ionic strength. This interaction had more influence on Qe than did the other interactions (temperature–pH, temperature–ionic strength). The finding exhibits 23 mg/g of desorption efficiency at 60 °C for 100 mg/L and 150 min of contact time.

**Author Contributions:** Conceptualization, M. and S.I.; Methodology, M. and S.I.; Supervision, S.I.; Writing-Original Draft Preparation, M.; Writing-Review & Editing, M.R. Formal analysis, A.A.

**Funding:** The authors acknowledge Fundamental Research Grant Scheme, FRGS (203/PJKIMIA/6071413), Research University Grant, RUI (1001/ PJKIMIA/ 814269), USM fellowship and Kementerian Pengajian Tinggi Malaysia for providing the financial support during Master.

**Conflicts of Interest:** The authors declare no conflict of interest.

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
