# Peer review of "Optimization Study for the Desorption of Methylene Blue Dye from Clay Based Adsorbent Coating"

_water, doi:10.3390/w11061304_

Round 1
Reviewer 1 Report
Dear authors:
English needs to be improved. At times it is difficult to understand what is written. There also seems to be some problem with superscripts and subscripts, including the "ºC" symbol.
-Information on methods is incomplete in some respects. For example: a) What is the binder made of? b) How many distilled water do you use? (both in paragraph 2.2)
-In equation (4), WAA and WAB are defined as the same concept. Also, equations (3), (4) and (5) must be better explained in the text.
-Line 122 in page 3. β11, β22 and β33 are also model coefficients.
-Lines 161-162 in page 4. "...The maximum desorption efficiency of 60 % was 161 attained at acid to alcohol ratio of 75:25% for thermo-chem method...". The maximun is for HCl.
-In table 5, predictor x2-x3 p-value is >>0.05. It is an error into the table? Or it must be removed from equation (7)?
-Line 236. "Student" was the pseudonym of the chemist William Sealy Gosset, and must be written in capital letters.
Author Response
Comments and Suggestions for Authors
Dear authors:
English needs to be improved. At times it is difficult to understand what is written. There also seems to be some problem with superscripts and subscripts, including the "ºC" symbol.
Response – As advised by the reviewer, English has been improved; superscripts and subscripts problem also has been corrected.
-Information on methods is incomplete in some respects. For example: a) What is the binder made of? b) How many distilled water do you use? (both in paragraph 2.2)
Response –
(a) Sentence added and highlighted by light blue “Water based binder was used as a support which contains Polyvinyl alcohol (PVA), ammonia, calcium carbonate, formalin and hydrosol. The contents present in binder are similar to bentonite clay except SiO2 which present in half amount to that of SiO2 present in bentonite. The percentage of calcium carbonate is higher in binder and also considered as major content [15].” From line no 76-80.
(b) 3g of distilled water was used.
-In equation (4), WAA and WAB are defined as the same concept. Also, equations (3), (4) and (5) must be better explained in the text.
Response – As advised by the reviewer, sentence has been corrected as “WAA is weight after adsorption, WAB is weight before adsorption”.
-Line 122 in page 3. β11, β22 and β33 are also model coefficients.
Response - As advised by the reviewer, sentence has been corrected.
-Lines 161-162 in page 4. "...The maximum desorption efficiency of 60 % was 161 attained at acid to alcohol ratio of 75:25% for thermo-chem method...". The maximun is for HCl.
Response – As advised by the reviewer, sentence has been corrected.
-In table 5, predictor x2-x3 p-value is >>0.05. It is an error into the table? Or it must be removed from equation (7)?
Response –p-value for x2-x3 is 0.011. It has been corrected in Table.
-Line 236. "Student" was the pseudonym of the chemist William Sealy Gosset, and must be written in capital letters.
Response - As advised by the reviewer, sentence has been corrected.

Reviewer 2 Report
The paper concerns the desorption study of methylene blue dye from bentonite adsorbent. Authors used combined thermal and chemical treatment of the adsorbent. Additionally, significant part of the manuscript is related to optimization study in order to define the optimal conditions of desorption process. Although the paper consist mostly with mathematical optimization its scope is interesting. However, the scientific novelty should be underlined more clearly. The introduction section is very comprehensive and it is directly related to the manuscript scope. Some arguable elements are in the experimental section.
There is no information what was the efficiency of MB adsorption – in other words what was the amount of dye adsorbed per unit mas of the adsorbent – this information would allowed to discuss desorption efficiency more clearly.
What were the optimal conditions of MB dye adsorption in order to achieve the maximum adsorption efficiency?
What was the maximum adsorption capacity of bentonite towards MB dye?
Authors used two different solvents to regenerate the adsorbent – did the Authors observe the decomposition of the bentonite after thermal and chemical treatment? Chemical composition of the adsorbent before and after such treatment should be presented. Additionally, what were the parameters of the porous structure of initial bentonite and how they change after regeneration tests?
In order to present thermal stability it would be better to present TG/DTA curves instead of table 2.
What do You mean by “desorption capacity”? Capacity is commonly used when we want to underline the amount of adsorbate that can be loaded to the selected adsorbent.
Authors state that they use bentonite without any modification. In section 2.2. there is information concerning binder – what was its role and what type of binder do the authors used?
What do the Authors mean by “adsorbent coating” – more detailed description should be provided.
The statement: “The finding exhibit maximum 20 desorption capacity 23 mg/g at 60oC for 100 mg/L of dye and 150 min of contact time” need to be rewritten to avoid misunderstanding.
What is an advantage of response surface methodology in such experiments? How the results obtained can affect the application of such adsorbent/adsorbate systemin real wastewater conditions? Is it possible to use such experiments to predict how many adsorption/desorption cycles will the bentonite work effectively.
In order to increase the scientific value of the manuscript it is suggested to present the research scope or mechanism of interaction (desorption) graphically.
Author Response
Comments and Suggestions for Authors
The paper concerns the desorption study of methylene blue dye from bentonite adsorbent. Authors used combined thermal and chemical treatment of the adsorbent. Additionally, significant part of the manuscript is related to optimization study in order to define the optimal conditions of desorption process. Although the paper consist mostly with mathematical optimization its scope is interesting. However, the scientific novelty should be underlined more clearly. The introduction section is very comprehensive and it is directly related to the manuscript scope. Some arguable elements are in the experimental section.
- There is no information what was the efficiency of MB adsorption – in other words what was the amount of dye adsorbed per unit mas of the adsorbent – this information would allowed to discuss desorption efficiency more clearly.
Response – The amount of MB adsorbed on adsorbent coating is 213.57 mg/g and is added in introduction section (from line 45-48, highlighted with yellow colour)
- What were the optimal conditions of MB dye adsorption in order to achieve the maximum adsorption efficiency?
Response- The adsorption of MB on adsorbent coating was carried out by Azha et al., (2015). The adsorption efficiency of MB was 99.9 % at room temperature (30oC) for about 6-7 hour of time with 100 mg/L of initial dye concentration. There was no effect of pH on adsorption process. Moreover, fastest or slowest mixing speed did not affect the adsorption process.
- What was the maximum adsorption capacity of bentonite towards MB dye?
Response –From the comparison data collected by Rafatullah et al. (2010) in his review study on adsorption of Methylene blue from low-cost adsorbent, the adsorption capacity of bentonite is 175 mg/g.
[Rafatullah, M., Sulaiman, O., Hashim, R., & Ahmad, A. (2010). Adsorption of methylene blue on low-cost adsorbents: a review. Journal of Hazardous Materials, 177, p. 70–80.]
- Authors used two different solvents to regenerate the adsorbent – did the Authors observe the decomposition of the bentonite after thermal and chemical treatment? Chemical composition of the adsorbent before and after such treatment should be presented. Additionally, what were the parameters of the porous structure of initial bentonite and how they change after regeneration tests?
Response – The adsorbent is basically a mixture of bentonite, water-based binder (which includes Polyvinyl alcohol (PVA), ammonia, calcium carbonate, formalin and hydrosol) and water. The prepared slurry was coated on glass beaker, where, binder act as a support and helps in coating the adsorbent layer on glass beaker. Therefore, it is difficult to separate bentonite from such mixture. However, according to FTIR results, the groups (Al-Al-OH and Al-Mg-OH groups) of bentonite does not change before and after regeneration which shows that bentonite does not affected. Those characterization are already present in another paper which is under review.
- In order to present thermal stability it would be better to present TG/DTA curves instead of table 2.
Response – As advised by reviewer the following correction has been done. Figure 1 added at Page No. 4.
- What do You mean by “desorption capacity”? Capacity is commonly used when we want to underline the amount of adsorbate that can be loaded to the selected adsorbent.
Response – Desorption capacity means the amount of dye desorbed from per gram of dye saturated sorbent at equilibrium. Sentence added from line 109-110.
- Authors state that they use bentonite without any modification. In section 2.2. there is information concerning binder – what was its role and what type of binder do the authors used?
Response – Binder is basically a water-based binder which is used as a support and helps in coating the adsorbent layer on glass beaker. According to Azha et al., (2015), 3 types of binder were used; water-based, oil-based and latex based binder. The water and latex based binders gave better mixability with clay-based adsorbent in the presence of distilled water as solvent. However, clay-based adsorbent and water as solvent cannot be dissolved in oil-based binder. The adsorbent coating of water-based binder gave the best and fastest adsorption compared to latex and oil-based binder. Within the first 2 hours, the adsorption reached 100 % removal.
[S.F. Azha, A.L. Ahmad & S. Ismail, (2015), “A New Approach of Thin Coated Adsorbent Layer for Batch Adsorption Using Basic Dye”, AJChE, 15, p. 10-21]
- What do the Authors mean by “adsorbent coating” – more detailed description should be provided.
Response – Description about adsorbent coating is given in introduction from line 41-53 “An adsorbent coating is a new approach of adsorbent where the ordinary form of adsorbent has been reformulating and transform into a liquid/slurry form which then been laminated onto an inert surfaces or substrate. Bentonite coating is an efficient methodology not only for durability and low cost, but also due to its acceptability in industries. Thin coated bentonite-based adsorbent had been prepared by mixing bentonite, water-based binder, and solvent in a specific ratio to remove MB from synthetic dye solution. For few reasons, it is desirable to have the solid adsorbent deposited on a substrate as a coating instead of being contained in particulate form as pellets, beads, flakes, particles, powder etc. Adsorbent coatings improve the catalytic and adsorption capacity of adsorbents by increasing the surface area/weight ratio, reduces the quantity of solid adsorbent required, enhances the binding strength, protects the substrate from harmful environment, and performs a specific desorptive or catalytic role over the entire surface of the substrate”.
- The statement: “The finding exhibit maximum 20 desorption capacity 23 mg/g at 60oC for 100 mg/L of dye and 150 min of contact time” need to be rewritten to avoid misunderstanding.
Response - As advised by reviewer the sentence is corrected
- What is an advantage of response surface methodology in such experiments? How the results obtained can affect the application of such adsorbent/adsorbate systemin real wastewater conditions? Is it possible to use such experiments to predict how many adsorption/desorption cycles will the bentonite work effectively.
Response –
1) Response surface methodology is used to optimize the conditions for any process and to predict a relationship between inputs and outputs. It helps in prediction of possible results of the process under the given conditions. It minimizes the number of experiments which otherwise may go up to thousands and thousands of input and output combinations. Running a large number of experiment is highly expensive and also not feasible under general laboratory conditions.
2)RSM is used for the same purpose in these studies where adsorption/desorption of MB on bentonite coating (not only bentonite) was studied. The optimization of desorption involving solvent concentration, temperature and time of exposure of desorption of MB was optimized using RSM.
3) The model fitted very well and provides the clue of the mechanism of desorption process in this system.
4) Yes, RSM can be used for to predict number of adsorption/desorption cycles in an ideal process where chemical compositions of the adsorbent-adsorbate remains unchanged. However, it will be partially applicable if some change in chemical composition and hence surface interactions might change which may have not been included in the model.
5)It is a well known standard method in chemometry which chemist are rarely using but engineers are frequently using.
- In order to increase the scientific value of the manuscript it is suggested to present the research scope or mechanism of interaction (desorption) graphically.
Response – As advised by reviewer the following correction has been done. Figure 9 added at page no. 13.

Reviewer 3 Report
Dear Authors ,
After reading the corrected manuscript I found that the manuscript entitled “Optimization Study for the Desorption of Methylene Blue Dye from Clay Based Adsorbent Coating” (written by Momina, Mohd Rafatullah, Suzylawati Ismail, Anees Ahmad) is written at a very low scientific level. Actually it is an exact report of a laboratory exercise with a substantial description of the manual activities for the most proper preparation of the testing system, however without any scientific interpretation of the obtained results.
Author Response
Comments and Suggestions for Authors
Dear Authors ,
After reading the corrected manuscript I found that the manuscript entitled “Optimization Study for the Desorption of Methylene Blue Dye from Clay Based Adsorbent Coating” (written by Momina, Mohd Rafatullah, Suzylawati Ismail, Anees Ahmad) is written at a very low scientific level. Actually it is an exact report of a laboratory exercise with a substantial description of the manual activities for the most proper preparation of the testing system, however without any scientific interpretation of the obtained results.
Response-Thank you so much for your time, the level of the manuscript is now improved after removing all the comments given by other three reviewers, In fact few full desorption studies paper reported in the literature, so we carried out the combination of thermal and chemical regeneration techniques for the desorption process. Hopefully the revised manuscript will fulfil the requirement.

Reviewer 4 Report
This study deals with the problem of regeneration and desorption ability of materials. Some comments are given below:
1) Check Equations. Some "_" are presented!
2) Authors must highlight more the need of "good desorption"
3) References must be enriched with more in number updated works.
4) Apart from SEM images, some othe rcharacterizations will be useful (FTIR, XRD).
Author Response
Comments and Suggestions for Authors
This study deals with the problem of regeneration and desorption ability of materials. Some comments are given below:
1) Check Equations. Some "_" are presented!
Response – As advised by reviewer, all equations are corrected and highlighted in pink colour.
2) Authors must highlight more the need of "good desorption"
Response – As advised by reviewer, the few sentences were added from line 276-280 at page no. 12.
3) References must be enriched with more in number updated works.
Response – Corrected as per author guideline
4) Apart from SEM images, some other characterizations will be useful (FTIR, XRD).
Response-Other characterization are under study.

Round 2
Reviewer 1 Report
The article has been clearly improved and can be accepted for publication. But there are still some issues to correct:
-In the new text of lines 76-80, the sub-index of "SiO2" must be corrected.
-In line 103 (page 3) , QH i really equation 4, and QC, equation 5.
Author Response
The article has been clearly improved and can be accepted for publication. But there are still some issues to correct:
-In the new text of lines 76-80, the sub-index of "SiO2" must be corrected.
Response – As advised by reviewer, the correction has been done
-In line 103 (page 3) , QH i really equation 4, and QC, equation 5.
Response – As advised by reviewer, the correction has been done

Reviewer 2 Report
Although the Authors made response to all of my comments, I am not convinced about the adsorption efficiency of MB onto analyzed adsorbent. Did the Authors made the adsorption process by themselves or did they follow somebody procedure? From the response it is suggested that: The adsorption of MB on adsorbent coating was carried out by Azha et al., (2015).??? If so, it should be clearly stated in experimental section with a proper reference. Did the Authors verified those results before their experiments? What was the maximum adsorption capacity of Your material - not the material described in other references? Detailed response is needed here.
Additionally You can't state about the desorption capacity - the word "capacity" is closely related to the adsorption process and its efficiency. It describes the amount of the adsorbate that can be adsorbed per unit mass of adsorbent. In case of the desorption process You can state about "desorption efficency" or "desortpion rate". Please check the real meaning of the word "capacity" which do not match with the desorption process.
Author Response
Comments and Suggestions for Authors
- Although the Authors made response to all of my comments, I am not convinced about the adsorption efficiency of MB onto analyzed adsorbent. Did the Authors made the adsorption process by themselves or did they follow somebody procedure? From the response it is suggested that: The adsorption of MB on adsorbent coating was carried out by Azha et al., (2015).??? If so, it should be clearly stated in experimental section with a proper reference. Did the Authors verified those results before their experiments? What was the maximum adsorption capacity of Your material - not the material described in other references? Detailed response is needed here.
Response –
1) The concerns of reviewers are natural. However, we did desorption study of MB on adsorbent coating. Other researchers (Azha et al) in our lab (Chemical Engineering Integrated Research Space Lab, University Sains Malaysia (USM)) under one of the authors of this paper, Dr Suzylawati Ismail, introduced this material (Bentonite based adsorbent coating) and explored its adsorption efficiency. The work is already published in three international journals which are as follows.
· Azha, S.; Ahmad, A.; Ismail, S. Thin coated adsorbent layer: characteristics and performance study. Desalin. Water Treat. 2015, 55, 956-969.
· Azha, S.; Ahmad, A.; Ismail, S. A New Approach of Thin Coated Adsorbent Layer for Batch Adsorption Using Basic Dye. AJChe. 2015, 15, 10-21.
· Azha, S.; Ahmad, A.; Ismail, S. Coating paint for dyes removal: Performance and characteristic. Journal of Water Process Engineering. 2017, 15, 18-25.
Carrying forward that work on the same adsorption setup, we have done the desorption study under Dr Suzylawati Ismail. Hence, authors were convinced that verification of those results were not necessary.
2) The reference is mention in experimental section 2.2 and introduction section.
- Additionally You can't state about the desorption capacity - the word "capacity" is closely related to the adsorption process and its efficiency. It describes the amount of the adsorbate that can be adsorbed per unit mass of adsorbent. In case of the desorption process You can state about "desorption efficency" or "desortpion rate". Please check the real meaning of the word "capacity" which do not match with the desorption process.
Response – Agree with the reviewer, we changed the term in the revised manuscript

Reviewer 3 Report
The authors made an attempt to take into account some requirements of the reviewers in relation to their manuscript. Unfortunately, some important suggestions have been omitted, namely, they did not control of the pH changes and it is not unambiguously stated that all measurements were made at a constant temperature. In addition, the authors did not explain the significance of the β... coefficients (lines 124-127).
Author Response
The authors made an attempt to take into account some requirements of the reviewers in relation to their manuscript. Unfortunately, some important suggestions have been omitted, namely, they did not control of the pH changes and it is not unambiguously stated that all measurements were made at a constant temperature. In addition, the authors did not explain the significance of the β... coefficients (lines 124-127).
Response-
We have done all experiments systematically in the manner that you are expecting. We control both pH and temperature. In each set of experiment the pH was controlled by fixing the concentration of solvent (HCl). In this way effect of concentration (pH) was studied at one temperature (30oC). The above experiment was repeated at more 4 temperature (40, 50, 60 & 70oC). Thus, in this study we have used 5 solvents at each temperature for desorption of dye; HCl solution (4M), three mixture of solvents (acid: alcohol ratio) and pure ethanol. Thus, pH was indirectly control at each temperature. In the previous studies different acid, bases and mixture of solvents were used.
β coefficients are just constants of developed model obtained by fitting the data. It accounts for combined effect of concentration, temperature and contact time.
I hope I am able to answer your queries satisfactorily.

Reviewer 4 Report
All my comments of the initial submission have been correctly replied and included in the revised manuscript. The quality of this work has been drastically improved after revision and therefore I recommend its publication as it is.
Author Response
All my comments of the initial submission have been correctly replied and included in the revised manuscript. The quality of this work has been drastically improved after revision and therefore I recommend its publication as it is.
Response – thank you so much

Round 3
Reviewer 2 Report
Authors have explained all of my comments in details.
Author Response
Authors have explained all of my comments in details.
Response – Thank you so much

Reviewer 3 Report
Dear Authors ,
The explanation and interpretation of the Figures 4, 5 and 6 are still insufficient and very questionable. Their interpretation should be broadened and more in-depth.
Author Response
Dear Authors ,
The explanation and interpretation of the Figures 4, 5 and 6 are still insufficient and very questionable. Their interpretation should be broadened and more in-depth.
Response- The captions of figures 4, 5 and 6 have been improved for more clarity of explanations and interpretations given in the text.
Request - Though, in my opinion, the interpretations and explanation of texts has been more clear after improving the captions of the set figures but still if anything is lacking kindly help me by suggesting your opinions (more than one) so that the best can be opted which does not hinder the spirit of current order.
Thanks.
